# Using Ecosystem Service Flows to Inform Ecological Compensation: Theory & Application

**DOI:** 10.3390/ijerph17093340

**Published:** 2020-05-11

**Authors:** Xiaolong Gao, Binbin Huang, Ying Hou, Weihua Xu, Hua Zheng, Dongchun Ma, Zhiyun Ouyang

**Affiliations:** 1State Key Laboratory of Urban and Regional Ecology, Research Center for Eco-Environmental Sciences, Chinese Academy of Sciences, Beijing 100085, China; xlgao_st@rcees.ac.cn (X.G.); bbhuang_st@rcees.ac.cn (B.H.); yinghou@rcees.ac.cn (Y.H.); xuweihua@rcees.ac.cn (W.X.); zhenghua@rcees.ac.cn (H.Z.); 2University of Chinese Academy of Sciences, Beijing 100049, China; 3Beijing Water Science and Technology Institute, Beijing 100048, China; mdc@bwsti.com

**Keywords:** ecosystem service flows, ecosystem services, ecological compensation, beneficiary pays, Fuzhou City

## Abstract

Ecological compensation is a crucial policy instrument that realigns the benefits of stakeholders to the ecosystem service provision. However, the spatial disconnections between locations where ecosystem services produced and used are common. The supply and demand for ecosystem services are calculated to reflect the status of the districts or counties based on ecosystem service flows. The replacement cost methods provide necessary technical supports for the calculation of compensation funds. The realigning of compensation funds between service-benefiting areas and service-providing areas not only identifies the beneficiaries and suppliers but also realizes the connection between them, which may be a feasible methodology. Fuzhou City is the study area, and two ecosystem services of water conservation and soil retention were taken into consideration. The prioritized development zone, Linchuan, and the key agricultural production zones paid ecological compensation funds. Linchuan paid the highest, 5.76 billion yuan. The key ecological function zones and the key agricultural production zones received the ecological compensation funds, of which Yihuang obtained the highest, 1.66 billion yuan. The realigning of compensation funds between the service benefiting and providing areas addresses the trade-offs between ecosystem services, social development, and ecosystem protection. Embedding the ecosystem service flows into the ecological compensation mechanism can most truly realize the value of ecosystem services, achieve the “beneficiary pays” principle, and be conducive to regional sustainable development.

## 1. Introduction

Ecosystem services (ES) are the aspects of the ecosystem that are being utilized for human well-being [1]. This includes security, the basic material for the good life, health, social relationships, and freedoms of choices and actions [2,3,4]. However, many ES belong to pure public goods (nonrival and nonexclusive) that could not be provided by the market mechanism. Policy instruments have been proposed to correct market failure and contribute to the ES supply. Nowadays, ecological compensation (equivalent to Payment for Ecosystem Services, PES) attracts increasing interests as a comprehensive mechanism to translate external, non-market values of ES into real financial values working as incentives for local actors [5,6,7]. The more popular the ecological compensation concept, the higher the demand for an appropriate indication, quantification, and spatial localization methods [8]. Positive progress has been made on the ES valuation [9,10,11,12,13], which includes the distribution of rights among the stakeholders [14,15], the transaction mechanism design [14,16], and the effectiveness assessment [5,15,16,17,18,19]. However, the standards for ecological compensation still remain a debatable topic [20,21,22,23]. Theoretically, there are two payment standards: opportunity cost method and ecosystem service assessment [21]. The former method taking the damage caused and the beneficiary’s payment capacity into consideration is the mainstream choice, while the latter calculates the total value involving various aspects of ES in large-scale projects where multi-parties are involved [24]. Existing ecological compensation projects mainly face problems to be solved in two aspects because of the adopted ecological compensation standards. First, the payment is so low that participants are calling for a higher payment standard. The opportunity cost could not reflect the value of ES that people benefit from. Second, the spatial disconnection between locations where ES are produced and where they are used are common [25]. Most studies implicitly assume spatial and temporal connections between ecosystem service suppliers and beneficiaries, while the actual connections, i.e., ecosystem service flows, are poorly understood [8,25]. The first problem can be solved by calculating the value of several typical ES in the ecological compensation standard, and the supply and demand accounting of ES can clarify the direction of ecosystem service flow. However, most of the currently available spatial ecosystem service studies focus on ecosystem service supply [8,11,26,27], whereas the demand side has not been sufficiently considered. Therefore, the concept of balance between supply and demand was introduced into the ecological compensation standard. According to the characteristics of the specific ecosystem service flow, the compensation standard will be able to identify the beneficiaries and suppliers and help to play the incentive role of policy tools to achieve sustainable development of the region [28].

In this work, two ES of water conservation and soil retention were selected to assess the ES supply/demand and to determine compensation funds and distribution methods. Fuzhou City was taken as an example to conduct empirical research and to calculate the realigning of compensation funds. The ecological compensation standard based on the ecosystem service flows has an objective scope of application and can realize the connection between service-benefiting areas and service-providing areas.

## 2. Study Area

Fuzhou City (26°29′–28°30′ N, 115°35′–117°18′ E) is located in the eastern part of Jiangxi Province (Figure 1). The terrain of Fuzhou is high in the south and low in the north and gradually slopes toward the plain area of Poyang Lake. The landform is dominated by mountains, hills, and valleys. The climate in Fuzhou City remains humid, and a rainy monsoon climate hovers in the south, with abundant rainfall. The average annual temperature is 16.9 °C to 18.2 °C, with an average annual rainfall of about 1700 mm. Fuhe River, the second-largest river in Jiangxi Province, has a catchment area of 16,800 square kilometers, which accounts for 84.6% of the city’s total land area, and a multi-year average runoff of 6.48 billion m^3^ [29]. The composition of Fuzhou’s main functional zones is nationally represented. Fuzhou’s 11 districts or counties can be classified into prioritized development zones and restricted development zones (key agricultural production zones and key ecological function zones). Scattered nature reserves (wetland parks, important water sources, world cultural and natural heritage, etc.) are classified as prohibited development zones [30].

Fuzhou City has a total population of 3.99 million, the urbanization rate of which is 44.97%, and the per capita gross domestic production (GDP) is 27,734 yuan (US $3962; 7 yuan = 1 US$). This is below the national average of 50,022 yuan (US $7146) [31]. It has faced a trade-off between social development and ecosystem protection and has research value in the Yangtze River economic belt.

## 3. Materials and Methods

### 3.1. The Selection of ES

Ecological compensation is a policy solution for realigning private and social benefits that result from decisions related to the environment [6]. ES provides the bridging communication platform between human society and nature [32]. The selection of ES mainly takes four aspects into account (Table 1). First, ES is defined as the instrumental values of the ecosystem for human well-being. The distinction between intermediate services and final services may be made [33]. Some ES such as soil formation and nutrient cycling are contributors to the end of human well-being; however, other final services connect to benefits directly, just as shown in the cascade model [34]. So, the final services are the objects of our research. Second, ES can be classified into four categories according to their excludability and rivalness status [33,35,36]. Pure public goods are the target of ecological compensation, because their natural capital and scarcity cannot be achieved through the market mechanism [37]. Obviously, most regulating and cultural services are public goods [33]. Third, the spatial characteristics of ecosystem service flows describe the relationships between the service-benefiting areas and service-providing areas [1,8]. Compared to in-situ and decoupled ecosystem service flows, directional and omni-directional ecosystem service flows involve conflicts of interest between beneficiaries and suppliers. Compared to omni-directional ecosystem service flows, the characteristics, service-providing areas, and benefiting areas of directional ecosystem service flows are easier to be identified, such as storm regulation, soil retention, water conservation, etc., especially at the watershed scale [25]. Fourth, the match between supply and demand of ES involves data from multiple fields, such as remote sensing and statistics. There are few studies in Fuzhou City, and public access to data is tough, such as the regional pollutant discharge data. The research finally selects water conservation and soil retention to realize the connection between service-benefiting areas and service-providing areas, match the supply and demand, and explore the applicable ecological compensation standards.

### 3.2. Data Sources

The land use/land cover (LULC) was obtained from the remote sensing survey and assessment project of the Chinese Decade of Ecological Environment (2000–2010) and from the five-year changes of the ecological environment (2010–2015), which is taken from the Chinese Academy of Sciences and the Ministry of Ecology and Environment of the People’s Republic of China. Object-oriented multi-scale segmentation and the establishment of a decision tree were used for classification. This was achieved by using the United States Landsat remote-sensing image as a data source with the support of a classical sample database constructed from a large number of ground survey samples. The accuracy of the data was verified and maintained above 95% with a spatial resolution of 30 m (Table 2).

The soil map was provided from the Chinese soil database, which was adapted from the Harmonized World Soil Database. The source data were in the ratio of 1:1,000,000 provided by the Institute of Soil Science, Chinese Academy of Sciences in the second national land survey in 1995 [38].

Daily climate data were obtained from the Resources and Environmental Sciences Data Center, Chinese Academy of Sciences [39]. ArcGIS software was used to calculate the inverse distance weighted spatial interpolation to form a 2000–2015 annual average temperature and annual precipitation spatial dataset. 

Shuttle Radar Topography Mission (SRTM) was the digital elevation model (DEM) that was used in the study. These data produced by the National Aeronautics and Space Administration (NASA) had a spatial resolution of 90 m. The slope data were generated from the DEM by the slope analysis tool in ArcGIS Desktop 10.2. 

Fuzhou Statistical Yearbook (2016) provided the socioeconomic data for 2015, including population, area, type and quantity of livestock, crop type and area, and GDP.

### 3.3. Assessment of Ecological Compensation Funds 

#### 3.3.1. Valuation and Mapping of ES Supply and Demand

##### Water Conservation

Water conservation supply. Water conservation is taken as the supply indicator for the regional ecosystem’s water conservation functions [40]. The water balance equation is used to calculate the water conservation, which is mainly related to factors such as precipitation, evapotranspiration, surface runoff, and vegetation cover types [41].
(1)$$S_{wc}=\sum_{i=1}^{j}\left(P_{i}-R_{i}-{ET}_{i}\right)\times A_{i}$$
where *S_wc_* is the total water conservation (m^3^), *P_i_* is rainfall (mm), *R_i_* is surface runoff (mm), *ET_i_* represents evapotranspiration (mm), *A_i_* is the area of type *i* ecosystem (km^2^), *i* is the ecosystem type in the study area, and *j* is the number of ecosystem types taken in the study area. 

Surface runoff is obtained by multiplying the rainfall by the surface runoff coefficient. Surface runoff coefficient refers to the ratio of surface runoff (mm) to rainfall, which reflects the capacity of the ecosystem water conservation to some contents. The surface runoff coefficient is obtained by consulting the literature, mainly including publicly available literature on precipitation and surface runoff data for various types of ecosystems [41]. The model had been validated further in the assessment of China and the Yangtze River Basin [11,42], which was considered reasonable.

Water conservation demand. Water resource demand (*D_wc_*) is the sum of the human water demand and river ecological flow requirements in a specific area [4]. We use human and environmental water requirements to represent the demand for water resource provisions. Human water requirements include water consumption by livestock, residents, and agricultural (e.g., irrigation) and industrial water [40]. River ecological flow requirements refer to the average flow that a river should maintain in order to sustain a good ecological function, which accounts for 30% of the average annual discharge [4,43].
(2)$$D_{wc}=P\times Q_{p}+\sum_{i=1}^{n}\left(L_{i}\times Q_{i}\right)+\sum_{i=1}^{m}\left(I_{j}\times Q_{j}\right)+G_{in}\times Q_{in}+F\times 30\%$$
where *D_wc_* is the demand for water resources (m^3^), *P* is the total population, *Q_p_* is the quota for rural and urban residents’ water consumptions (m^3^·capita^−1^), *L_i_* is the number of type *i* livestock, and *Q_i_* is the water quota for each type of livestock. *I* is the planting area (ha) of crop type *j*, and *Q_j_* is the water quota for crop type *j* (m^3^·ha^−1^). *G_in_* is the industrial added value, and *Q_in_* is the water consumption of the industrial added value of ten-thousand yuan. *F*, the multi-year average runoff (m^3^), refers to Fuzhou Water Resources Bulletin 2015 [29]. The quotas of water use for the population, livestock, and crops are based on Jiangxi standards [44,45,46].

##### Soil Retention

Soil retention supply. Soil retention supply (*S_sr_*) refers to the ability of an ecosystem to maintain soil for a certain period of time and is characterized by the inverse variable of soil erosion intensity (t·ha^−1^·yr^−1^). Soil and Water Assessment Tool (SWAT) was used to estimate soil yield by the modified universal soil loss equation (USLE) [47,48].
(3)$$S_{sr}=R\times K\times LS\times C$$
where *R*, *K*, *LS*, and *C* denote the rainfall erosivity factor (MJ·mm·ha^−1^·h^−1^·yr^−1^), soil erodibility factor (t·ha·h·ha^−1^·MJ^−1^·mm^−1^), topographic factor, and crop/vegetation cover factor, respectively.

Soil retention is an estimate rather than an actual measurement of soil loss. In order to validate the regional application of the model, we validated the USLE model using soil erosion rates, which were estimates based on observation data from seven major watersheds in China [47,48]. The simulations (USLE) of soil erosion rates were very similar to the corresponding estimates (*R*^2^ = 0.657, *N* = 7, *p* < 0.05). Hence, we can confirm that USLE and its parameters can simulate general soil erosion in China. We then further validated the USLE model in the Yangtze River Basin using the area of soil erosion. Eleven provinces were selected as the main water erosion areas to compare our research with the results of the first China Water Census in 2013. The validation results showed that our estimates of soil erosion in these provinces were consistent with the census results (*R*^2^ = 0.977, *N* = 11, *p* < 0.01). Based on the above verification, the soil retention service model is considered to be reasonable [18]. 

Soil retention demand. The allowable amount of soil loss was taken as the demand indicator (*D_sr_*). The lower the soil erosion intensity, the stronger the soil retention effect. According to the standards for classification and gradation of soil erosion, the allowable amount of soil erosion in the southern red soil hilly areas are divided into 6 levels [49]. Since the intensity of soil loss in Fuzhou City is higher than the level of the micro-degree, the demand standard is set at 20 t·ha^−1^·year^−1^, which is mild. As the erosion intensity standard decreases, the quantity and proportion of payments in the study area will also change, but there will be no interference with the payment relationship. 

#### 3.3.2. The Matching between ES Supply and Demand

Most ES are public goods, and it is difficult to define their property rights [50]. The introduction of ecosystem service flows fulfills the connection between the service-benefiting areas and service-providing areas [8,25]. The watershed is a typical directional flow that further strengthened the relationship between administrative regions. Of Fuzhou’s administrative region, 84.6% is within the Fuhe River Basin. The net service-providing areas are the source areas, while the middle and lower reaches are both service-providing areas and benefiting areas. Different from other techniques (supply-demand ratio, proportional contribution of ES supply to demand, etc.) [4], the differences between the supply and demand of water conservation and soil retention can truly reflect the status of ES in districts or counties. This provides a basis for the setting of ecological compensation indicators.

##### Water Conservation

*G_wc_* represents the difference between water resources supply and demand.
*G_wc_* = *S_wc_* − *D_wc_*(4)

*G_wc_* reflects the water conservation status of districts or counties. If *G_wc_* is less than zero, the water conservation will be in deficit, and transfer payments will be required. If *G_wc_* is greater than zero, there will be a surplus of ecosystem service, and compensation will be available. The districts or counties will achieve a balance between supply and demand when *G_wc_* is equal to zero. 

##### Soil Retention

Since the soil erosion intensity is an inverse variable [4], *G_sr_* represents the difference between soil retention supply and demand.
*G_sr_* = *D_sr_* − *S_sr_*(5)

*G_sr_* represents the soil retention status of districts or counties. If *G_sr_* is less than zero, the ecosystem service will be in deficit, and transfer payments will be required. If *G_sr_* is greater than zero, there will be a surplus in the ecosystem service, and compensation will be available. If *G_sr_* is equal to zero, there is a balance between supply and demand. 

#### 3.3.3. Accounting for Ecological Compensation Funds 

Compensation funds are monetized in accordance with the replacement cost methods [51,52]. Districts or counties that have deficits in water conservation or soil retention pay compensation funds accordingly.

##### Water Conservation

The compensation funds for water conservation (*V_wc_*) are used to calculate the construction and the operation costs of the reservoir according to the replacement engineering approach [53].
*V_wc_* = *G_wc_* × (*C_re_* + *C_ro_*)(6)
where *C_re_* is the engineering cost per unit capacity of the reservoir, and *C_ro_* is the annual operating cost per unit capacity of the reservoir [54].

##### Soil Retention

The compensation funds for soil retention are calculated according to the replacement cost method [53], mainly for the treatment of sediment deposition and non-point source pollution.
*V_sr_* = *V_sd_* + *V_dpd_*(7)
*V_sd_* = *λ* × (*G_sr_*/*ρ*) × *c*(8)
(9)$$V_{dpd}=\sum_{i=1}^{n}G_{sr}\times c_{i}\times p_{i}$$

*V_sr_* stands for the compensation for soil erosion, *V_sd_* represents the cost of reducing the sediment deposition, *V_dpd_* represents the cost of reducing non-point source pollution, *λ* stands for the sediment deposition coefficient, *ρ* stands for the soil bulk density (t/m^3^), *c* stands for the cost of the unit reservoir dredging project (yuan/m^3^), *c_i_* represents the type *i* pollutants (e.g., nitrogen and phosphorous) in the soil, and *p_i_* stands for the unit treatment cost of type *i* pollutants [54].

#### 3.3.4. Realigning of Ecological Compensation Funds

ES are external [1,55], especially at the Fuhe River Basin scale. This has resulted in the spillover of ES provided by Fuzhou City to other administrative areas. Therefore, the allocation of compensation funds paid by ecological deficit areas depends on the surplus ratio (*SR_i_*) of ecological surplus areas. The sole ecological surplus area can get all compensation funds. The “beneficiary pays” principle is realized through the realigning of funds [56], which builds the connection between service-benefiting areas and service-providing areas.
(10)$${SR}_{i}=(G_{i}/\sum_{i=1}^{n}G_{i})\times 100$$
where *G_i_* represents the difference between the ES supply and demand in ecological surplus areas.

## 4. Results

### 4.1. Water Conservation

As depicted in Table 3 and Figure 2a, the prioritized development zone (Linchuan) and the key agricultural production zones (Dongxiang, Jinxi, and Chongren) fall in the middle and lower reaches of the Fuhe River. Since these are water conservation deficit districts or counties, they need to pay for ecological compensation funds. Among them, Linchuan paid the highest amount (5.75 billion yuan).

The five key ecological function zones (Nanfeng, Yihuang, Zixi, Lichuan, and Guangchang) and the key agricultural production zones (Le’an and Nancheng) in the upper reaches of the river belong to the ecological surplus areas in water conservation, and ecological compensation funds can be obtained. Yihuang’s surplus ratio reached 18.97%, obtaining 1.74 billion yuan.

### 4.2. Soil Retention

The key ecological function zones in Fuzhou City belongs to the ecological function zone of soil and water conservation in the Wuyi Mountains [30]. As shown in Table 4 and Figure 2b, Fuzhou City faces a serious soil erosion problem, and the upstream soil loss is in grave condition. Except for Dongxiang, the remaining 10 districts or counties all belong to ecological deficit areas in soil retention. The sediment and pollutants carried by the Fuhe River put pressure on river management and water quality improvement in the middle and lower reaches. Guangchang, the source of the Fuhe River, paid the highest compensation funds of 98.81 million yuan. Dongxiang, located downstream, received a total of 557.03 million yuan for compensation.

### 4.3. Realigning of Ecological Compensation Funds

The two ES of water conservation and soil retention are comprehensively measured, and the assessment results are shown in Table 5 and Figure 3. The prioritized development zone (Linchuan) and the key agricultural production zones (Dongxiang, Jinxi, and Chongren) were required to pay for ecological compensation funds. Linchuan paid the highest (5.76 billion yuan). The key ecological function zones (Nanfeng, Yihuang, Zixi, Lichuan, and Guangchang) and the key agricultural production zones (Nancheng and Le’an) received the ecological compensation funds, of which, Yihuang obtained the highest (1.66 billion yuan).

According to the realigning of ecological compensation funds, suppliers and beneficiaries can be preliminarily determined. As shown in Figure 3, Linchuan, Dongxiang, Jinxi, and Chongren in the lower reaches of the Fuhe River can be regarded as service-benefiting areas, while the other districts or counties in the upper and middle reaches are regarded as service-providing areas. Figure 3 basically reflects the direction of ecosystem service flows.

## 5. Discussion

### 5.1. The Practical Needs for Ecological Compensation Standards Based on Ecosystem Service Flows

According to the source of ecological compensation funds, projects are mainly divided into government-financed projects and user-financed projects [5]. Since most regulating and cultural services are public goods [33], the existence of externalities has led to free-riding [37]. The government, acting on the behalf of users, conservation institutions, has become the crucial force of ecological compensation. Many large-scale ecological compensation projects in various countries are led by the government, such as Natural Forest Protection Engineering and Grain-for-Green Program in China, Conservation Reserve Program in the USA, the Payments for Environmental Services program in Costa Rica, and the Payments for Hydrological Environmental Services Program in Mexico [5,14]. At present, China promotes ecological compensation at a watershed scale [14]. Since the relationship is between the upper and lower reaches of the river basin, rights and obligations are easier to define. Services such as water conservation, soil retention, and water regulation can be used as indicators of ecological compensation. In addition, in the advancement of ecological civilization, China has put more emphasis on market-based compensation methods [37]. Therefore, a standard system that can reflect the value of ES and establish the connection between suppliers and beneficiaries needs to be found [28]. 

### 5.2. Addressing Double-Trade-offs Between ES, Social Development, and Ecosystem Protection

Among the directional ecosystem service flows established on the Fuhe River Basin, the upstream provides benefits of water resources to the middle and lower reaches, while the soil erosion in the upstream increases the risk of siltation and water quality deterioration in the middle and lower reaches. It reflects the trade-off between ES at the watershed scale. To balance the interests of the upstream/downstream and the left/right banks, relevant ES need to be considered. 

Eighty-four point six percent of Fuzhou’s administrative districts are in the Fuhe River Basin [29], facing the trade-off between social development and ecosystem protection [37,57], which is also typical in the Yangtze River Basin. Restricted development zones are committed to ecosystem protection, which hinders industrial development [37]. The local government does not enjoy the benefits that should be obtained by ES provision. In addition, ecological restoration and protection require large amounts of capital investment, which becomes an additional financial burden [58]. There is a need for the new policy instruments that are urgently used to address conflicts of interest among stakeholders. 

Service-providing areas are usually underdeveloped areas [18]. Referring to the compensation standard based on ecosystem service flows, service-benefiting areas and service-providing areas can be identified, and the ecological compensation funds are also accounted for using the replacement cost methods. Therefore, the ES-facing market failures have value, and the payment ratio and requirements can be negotiated under the guidance of the higher-level governments or between the governments at the same level. 

ES are all the benefits humans get from ecosystems [1]. After the implement of the policy, service-benefiting areas will pay compensation funds to obtain continuous ES, such as water conservation and soil retention. Service-providing areas will use the compensation funds to commit to ecosystem protection investment, regional development planning, and residents’ livelihood improvements [16]. Through the internalization of externalities between regions, a mutually beneficial and win-win relationship is established, and regional sustainable development is promoted.

### 5.3. The Legitimacy of Ecological Compensation Policy 

The Chinese government proposed the construction of ecological civilization in 2012. Among them, deepening the reform of institutional mechanisms is the core of the work. As a basic system, the improvement of the natural resources asset property rights system is the focus of recent work, and ecological compensation in the watershed is one of the representative policies [14]. At present, various forms of horizontal ecological compensation policies for river basins have been launched across the country. As mentioned earlier, the implementation of ecological compensation policies is in line with the laws of natural development and reflects the value of ES. Since there is no national law that regulates this policy, it is currently more of a pilot project at the regional level and a decision to explore the path of sustainable development on its own. Therefore, it is necessary to conduct a scientific and legitimacy test on policies, standards, and the use of funds. Optimistically, the ecological compensation regulations have entered the legislative process, and related work will make greater progress in the future. 

### 5.4. The Limitations

The results of compensation fund allocations are basically consistent with the national plan for the major functional zones. Key ecological function zones receive compensation benefits by providing ES. Obviously, as more ES are incorporated into the standard system, the relationship between participants is more in line with reality. However, the funds for ecological compensation have also increased accordingly, exceeding the participants’ ability to pay. This is still not the major problem, and the study has proposed that the pressure on local governments to pay compensation funds should be alleviated by multi-level government prorating [21]. It is important that the compensation standard system proposed based on a watershed scale is not applicable to most ES, because for omnidirectional flows, such as pollination, climate regulation, and carbon sequestration [1,33], the research on beneficiaries and suppliers is still insufficient [25,59]. Therefore, an ecological compensation standard based on ecosystem service flows is still only applicable to basin scales with the characteristics of directional service flows. More efforts need to be devoted to ecosystem service flow profiling and quantitative research [8,25]. In addition, Fuzhou City has already piloted watershed ecological compensation using the water quality and quantity of river sections as indicators in 2019. The environmental and socioeconomic effects of the pilot method need to be comprehensively evaluated [18]. The results may be beneficial to the ecological compensation based on the ecosystem service flows. So, the new method system needs further verification.

## 6. Conclusions

The theory of ecosystem service flows is introduced into the ecological compensation mechanism, which can truly reflect the real status of the study areas by calculating the supply and demand. The replacement cost methods provide necessary technical support for the calculation of compensation funds. The realigning of compensation funds between service-benefiting areas and service-providing areas not only identifies the beneficiaries and suppliers but also realizes the connection between them, which may be a feasible methodology. As a study area, Fuzhou City has the political advantages and typical geographical advantages as an advanced demonstration zone of ecological civilization in Jiangxi Province, China. Fuzhou City is the epitome of the Yangtze River Basin, and it also faces the double-trade-offs between ES, social development, and ecosystem protection. The two ES of water conservation and soil retention can not only be used as compensation standards in Fuzhou City but can also be applied to the Yangtze River Basin. In addition, nutrient regulation is also indispensable. Under specific capital distribution policies and capital use policies, suppliers and beneficiaries will make decisions in line with green development, which will help promote regional sustainable development. The ecological compensation policy with ecosystem services as the accounting indicator is currently actively promoted by China and is an important content of the natural resources asset property rights system. In order to protect the interests of stakeholders, the ecological compensation policies, standards, and use of funds need to be reviewed scientifically and legally. In addition, more efforts are needed in ecosystem service flow profiling and quantitative research. 

## Figures and Tables

**Figure 1 ijerph-17-03340-f001:**
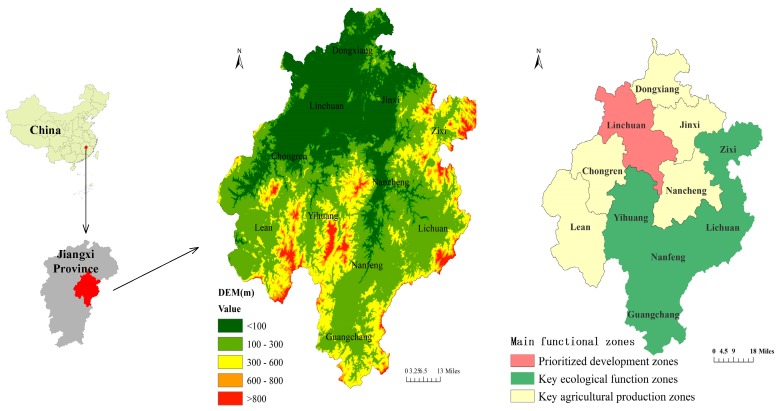
The research area of Fuzhou City located in the eastern part of Jiangxi Province covers a total area of 18,800 square kilometers. The main functional zones are classified as prioritized development zones (Linchuan); key agricultural production zones (Dongxiang, Jinxi, Nancheng, Chongren, and Le’an); and key ecological function zones (Yihuang, Nanfeng, Guangchang, Lichuan, and Zixi).

**Figure 2 ijerph-17-03340-f002:**
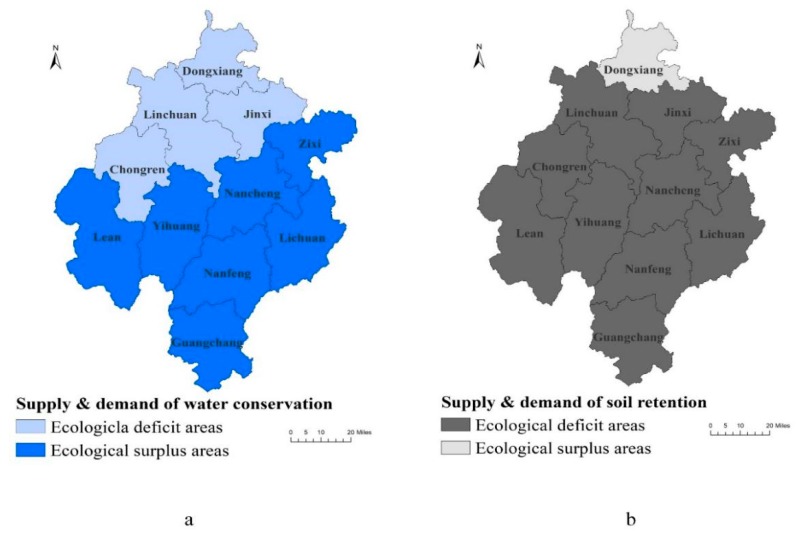
Supply and demand for water conservation and soil retention. (**a**) Districts or counties can be classified into ecological deficit areas and ecological surplus areas by calculating the difference between supply and demand for water conservation. Dongxiang, Linchuan, Jinxi, and Chongren belong to ecological deficit areas in water conservation. (**b**) Dongxiang is the sole ecological surplus area in soil retention.

**Figure 3 ijerph-17-03340-f003:**
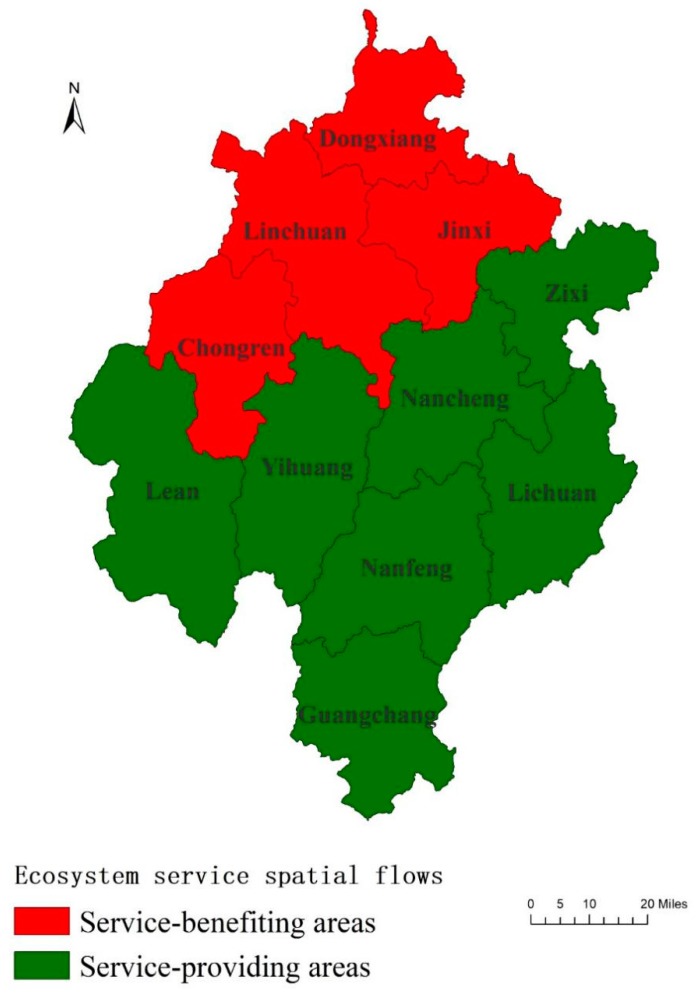
Identification of service-benefiting areas and service-providing areas. Districts or counties can be classified into service-benefiting areas or service-providing areas. The classification basically reflects the spatial direction of the ecosystem service flows.

**Table 1 ijerph-17-03340-t001:** The selection of the ecological compensation (ES).

Type of Services	Service	Final Services	Pure Public Goods	Spatial Characteristics	Final Selection
Provisioning services	Food	Yes	No	Decoupled flow	No
	Genetic resources	Yes	Yes	Decoupled flow	No
	Fresh water	Yes	No	Directional flow	No
Regulating services	Water quality regulation	Yes	Yes	Directional flow	No
	Climate regulation	Yes	Yes	Omni-directional flow	No
	Water conservation	Yes	Yes	Directional flow	Yes
	Soil retention	Yes	Yes	Directional flow	Yes
	Pollination	Yes	Yes	In situ	No
Cultural services	Recreation and ecotourism	Yes	No	Decoupled flow	No
	Aesthetic values	Yes	No	Decoupled flow	No
Supporting services	Soil formation	No	Yes	In situ	No
	Primary production	No	Yes	In situ	No
	Nutrient cycling	No	Yes	In situ	No

**Table 2 ijerph-17-03340-t002:** Sources of principal data.

Data	Data Resolution	Data Source
Land Use/Land Cover (LULC)	30 m	Remote sensing survey and assessment project of the Chinese Decade of Ecological Environment (2000–2010) and the five-year changes of the ecological environment (2010–2015)
Soil map	— ^1^	Harmonized World Soil Database version 1.2
Daily climate data	—	Resource and Environment Data Cloud Platform, Data Center for Resources and Environmental Sciences, Chinese Academy of Sciences
Digital Elevation Model (DEM) data	90 m	National Aeronautics and Space Administration (NASA)
Socioeconomic data	county scale	Statistical Yearbook

^1^ “—” indicates no value.

**Table 3 ijerph-17-03340-t003:** Assessment of supply and demand for water conservation and the distribution of compensation funds.

Item	Linchuan	Nancheng	Dongxiang	Jinxi	Chongren	Le’an	Nanfeng	Yihuang	Zixi	Lichuan	Guangchang
*S_wr_* (billion m^3^)	1.13	1.35	0.74	0.89	0.97	1.66	1.56	1.64	1.18	1.45	1.20
*D_wr_* (billion m^3^)	1.84	1.00	1.03	1.01	0.98	1.20	0.97	0.89	0.51	0.85	0.68
Compensation payment (billion yuan)	5.75	— ^1^	2.35	0.97	0.09	—	—	—	—	—	—
Compensation funds received (billion yuan)	—	0.81	—	—	—	1.07	1.38	1.74	1.56	1.39	1.21
Surplus ratio (%)	—	8.87	—	—	—	11.66	15.07	18.97	17.04	15.19	13.19

^1^ “—” indicates no value. *S_wr_*: the total water conservation (m^3^). *D_wr_*: the demand for water resources (m^3^).

**Table 4 ijerph-17-03340-t004:** Assessment of supply and demand for soil retention and the distribution of compensation funds.

Item	Linchuan	Nancheng	Dongxiang	Jinxi	Chongren	Le’an	Nanfeng	Yihuang	Zixi	Lichuan	Guangchang
*S_sr_* (million t)	5.00	6.00	2.00	3.00	4.00	12.00	17.00	10.00	5.00	6.00	11.00
*D_sr_* (million t)	4.25	3.43	2.54	2.71	3.04	4.82	3.83	3.87	2.50	3.42	3.21
Compensation payment (million yuan)	9.51	32.63	— ^1^	3.73	12.17	91.03	167.02	77.66	31.75	32.73	98.81
Compensation funds received (million yuan)	—	—	557.03	—	—	—	—	—	—	—	—

^1^ “—“ indicates no value. *S_sr_*: the ability of an ecosystem to maintain soil for a certain period of time (t). *D_sr_*: the allowable amount of soil loss (t).

**Table 5 ijerph-17-03340-t005:** The realigning of ecological compensation funds.

Item	Linchuan	Nancheng	Dongxiang	Jinxi	Chongren	Le’an	Nanfeng	Yihuang	Zixi	Lichuan	Guangchang
Compensation payment (billion yuan)	5.76	— ^1^	1.79	0.97	0.10	—	—	—	—	—	—
Compensation received (billion yuan)	—	0.78	—	—	—	0.98	1.21	1.66	1.53	1.36	1.11

^1^ “—” indicates no value.

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
