# Peer review of "Using Ecosystem Service Flows to Inform Ecological Compensation: Theory & Application"

_ijerph, 2020, doi:10.3390/ijerph17093340_

Round 1
Reviewer 1 Report
This is a manuscript of great interest since it deals with one of the most important issue concerning ES: the relation between supply and demand and their possibility to be compensated for the ES loss.
I don't have substantial remarks on this work, which is well-written and structured, with a straightforward methodology and a well-defined research area.
My main doubt regards 1) the better incorporation of the relation between ES and health otherwise it is not clear why this journal... 2) to highlight the relation between biophysical estimation and economic one and 3) the emphasis to policy implications (which actions with the payment).
You can find more comments on the attached file.
Good luck

Reviewer 2 Report
General comment
The paper is interested and in general the different sections are well structured, with the exception of some sub-sections. The authors pursue the objective of identifying a methodological path to promote realigning of compensation funds between the service benefiting and service providing areas in order to solve the issues on the trade-offs between ecosystem services, social development, and ecosystem protection.
Specific comments for sections
- The authors in section “1. Introduction” introduce the issue and set the objectives pursued by their study. This section, although as a whole, appears to be a coherent, needs more in-depth as the issue addressed is quite complex.
- The structure of the paragraphs and subparagraphs of section 2 appears a little confused, in fact I have found for this section a general title “2. Methodology and Material” to which follows the sub-section “2.1 Study area”, in which the authors introduce the area under study and to which it still follows a sub-section entitled “2.2 Material and Method”.
- The legends of Figure 1, which supports the description of the case study, do not seem consistent with the Figure capitation, so I recommend improving it.
- The sub-section “2.2.1 The selection of ES”, where the authors present the selected ES is unclear, in fact although they propose a structure according to first, second, third and fourth, the information is confused. The characterization of the four selected ES macro areas is not well structured and the detailed information on the individual ES is also not clear. I recommend to characterize the ES according to a general list and then to provide detailed and anyway it is necessary to improve this section.
- The statement reported in line 98-99, i.e. “So the final ES are the optimal objects” is not clear.
- I highlight a problem with the formatting of the formulas in sub-section 2.2.2, for example formula 1, 2 and 9.
- Authors should specify all acronyms as e.g. SWAT or SRTM DEM.
- I highlight an error in formula 6 where instead of "Gw" should be "Gwc".
- Section 3. Results is well structured.
- Section “4. Discussion” and in particular the two sub-sections “4.1. The practical needs for ecological compensation standard based on ecosystem service flows” and “4.2. Addressing double trade-off between ES, social development and ecosystem service” need a deepening. In fact, it is not clear how ecological compensation funds are distributed among the subjects, what types of interventions are implemented and how in this specific case the realignment takes place between the beneficiaries of the area and the producers for the services for the area.
- In particular, the position of the authors on the question of the legitimacy of benefiting from the ecosystem services produced by some areas, even if from the same region by means of the promise to pay for the service received, is not clear.
- The authors do not propose a reflection on the legitimacy of recovering funding also to conserve and develop ES flow in those areas that are dependent on those that produce them.
- At the moment the Section 5. Conclusions is currently a bit poor, but it could be improved on the basis of my suggestions proposed to integrate the Discussion section.
Round 2
Reviewer 2 Report
The authors have integrated the paper by solving the critical issues found in the previous version and therefore the paper can be considered suitable for publication.
I hope that in the future they have the opportunity to investigate the issue of ecological compensation by considering some aspects, which at the moment, although not clear set in the policies proposed by the Chinese government, are very important to promote sustainable and coordinated development, such as environmental and ecological equalization. The promotion of which must be supported by specific assessments and by the identification of a mechanism based on clear and well-structured rules, in order to guarantee the protection and conservation of environmental resources, intra-generational and intergenerational equity.